# Worldviews and the role of social values that underlie them

Rebekah Mifsud[1]*, Gordon Sammut[2]

1 Department of Cognitive Science, University of Malta, Msida, Malta, 2 Centre for the Study and Practice of Conflict Resolution, University of Malta, Valletta, Malta

* rebekah.mifsud@um.edu.mt

## Abstract

In today's ideologically diverse world, it is pertinent to have a better understanding of how our beliefs of the social world shape our thinking and behaviour. The purpose of this paper is to investigate the key social values that underlie particular sets of beliefs, referred to here as worldviews. Worldviews encompass beliefs that shape one's outlook on life and are, therefore, instrumental in providing meaning to one's reality and one's understanding as to how one fits in it. They can be classified into five unique types, namely, *Localised*, *Orthodox*, *Pragmatist*, *Reward*, and *Survivor*. In this paper we start by proposing a theoretical relationship between this five-factor typology and social values. Following this, we present findings that show that worldviews may be mapped onto the two higher order value dimensions of *Openness to Change versus Conservation*, and *Self-transcendence versus Self-Enhancement*. We conclude by outlining the implications that these findings have on understanding individual cognition and society in general.

## Introduction

Investigating beliefs naturally solicits the question of what purpose they serve. Beliefs exist at varying levels of generalizability and are shaped and reinforced by culture, experience, and theology [1]. For this reason, they serve multiple purposes drawing upon the need to form "enduring, unquestioned ontological representations of the world" [2]. When a set of related beliefs combine, they do so in terms of overlapping substantive content or shared functionality. Either way, when they do they form belief systems that, when coherently clustered, are recognisable as generalized worldviews or ideologies. The understanding of belief systems and how different beliefs bind together has been a relatively popular focus of social research [3–5]. From political ideologies to religious beliefs, various studies have suggested that belief systems need not just be contained within the individual but rather may also exist across individuals [6], facilitating ways of developing alliances with others, maintaining a shared reality, and extending the lifespan of the belief system beyond the believers themselves [1, 3]. It follows, therefore, that belief systems have an important role in both personal identity and society, serving the psychological needs of the individual as well as the institutionalised power structures of society. More relevant to our study is the need to understand how elements of belief systems, such as values [1], play a role in shaping and informing our understanding of

**Competing interests:** The authors have declared that no competing interests exist.

ourselves and the world around us, and in what manner they serve to guide our actions. The need to investigate this is evident when considering the highly divided world we live in. For instance, religious beliefs and political ideologies are known to exert a significant influence on social cohesion [7]. Accordingly, it is not unreasonable to hypothesise that worldviews too may play such a role. Investigating the values that underlie our worldviews is an essential step towards understanding the motives and perspectives of individuals that enable effective communication and collaboration amongst diverse actors.

In this paper, we start by defining worldviews and values and follow with foraging a theoretical linkage between the two. We then proceed by reporting the findings of a study of this theoretical overture before we conclude with a discussion of implications and suggestions for future study. We conclude by asserting that the study of worldviews offers a pathway towards the understanding of coalitional and oppositional projects undertaken by social groups in everyday life.

## Worldviews

Beliefs and belief systems share several essential properties and features, namely: they vary in generalizability and strength; they may arise and be reinforced by experience, culture, society, philosophy, and theology; they are instrumental in helping us provide meaning of the world, ourselves and our place in society; and they also share a strong relationship with behaviour [1]. It is reasonable, therefore, to propose that we are wired for storing beliefs and using them to navigate the world around us. Buhagiar and Sammut [8] explain how beliefs serve an extended dual purpose *of* describing elements in our environment *for* the purpose of guiding action. Similar sentiments have been subsequently proposed by Power et al. [9] with regards to world-making. This involves worldviews, that is, a particular set of generalized beliefs that we use to describe ourselves and the world around us. Worldviews encompass beliefs that shape our outlook on life and they are pertinent in providing meaning to our reality and our understanding of how we fit within it [10]. In an extensive review of the literature on worldviews, Koltko-Rivera [10] has pointed out that the construct of worldviews has oftentimes been defined and named in a multitude of ways, from cultural and value orientations aimed at conceptualizing worldviews at a social level [11–13], to philosophical outlooks aimed at conceptualizing worldviews at an individual level [14]. More than a decade later, this scenario on worldviews remains largely the same, lacking a unified understanding of the concept. We attempt to remedy this pitfall in the present paper. Similar to Koltko-Rivera [10], we define worldviews as representations of the structure of how and what people think. We propose that their function lies in how they operate to enable subjects to adapt their responses to present ecological demands [15, 16]. As outlined by Sammut et al. [16], a number of theoretical constructs fall in line with this definition of worldviews that have employed different terms. Sammut et al. (2022) identify four theories that propose remarkably identical five-factor typologies, namely: (i) symbolic universes [17], (ii) social axioms [18], (iii) moral foundations [19], and (iv) deep stories [20]. It is worth noting that these typologies possess various similar features. Firstly, they do not solely focus on individual disposition but situate individual dispositions within the wider social sphere, tapping into psychological constructs that shape the way individuals interpret their social world. Secondly, they all serve the practical purpose of enabling individuals to adapt suitably to different situational demands. For instance, the symbolic universe, *Interpersonal Bond*; the moral foundation, *Loyalty/Betrayal*; and the deep story profile, *Team Player*, emphasize pro-social behaviour. Conversely, the symbolic universe, *Others' World*; the social axiom, *Social Cynicism*; the moral foundation, *Authority/Respect*; and the deep story profile, *Cowboy*, emphasize selfish behaviour. With consideration to these commonalities, Sammut et al. [16]

**Table 1. A syncretic conceptualisation of worldviews as proposed by Sammut et al [16].**

| Worldviews | Symbolic Universes | Social Axioms | Moral Frameworks | Deep Stories |
|---|---|---|---|---|
| *Localised* | Interpersonal Bond | Social Complexity | Loyalty/Betrayal | Team Player |
| *Orthodox* | Ordered Universe | Religiosity | Purity/Sanctity | Worshipper |
| *Pragmatist* | Niche of Belongingness | Fate Control | Fairness/Reciprocity | Rebel with a Cause |
| *Reward* | Caring Society | Reward for Application | Harm/Care | Cosmopolitan |
| *Survivor* | Others' World | Social Cynicism | Authority/Respect | Cowboy |

proposed the notion of worldviews, offering a novel five-factor typology aimed at unifying the above-mentioned concepts. The five worldview types include the (i) *Localised*, (ii) *Orthodox*, (iii) *Pragmatist*, (iv) *Reward*, and (v) *Survivor* worldviews [16]. As summarised in Table 1, each worldview captures a symbolic universe, social axiom, moral foundation, and a deep story profile. The *Localised* worldview involves the desire to fix problems or address social issues. The *Orthodox* worldview seeks to preserve the status quo. The *Pragmatist* worldview is protective and revolves around self-interest. The *Reward* worldview centres around determination to work hard to obtain a desired goal. Lastly, the *Survivor* worldview involves fatalism, distrust in others and the need to overcome adversity [16]. It is worth noting that what differentiates these worldviews from the other similar five-factor typologies is the way in which they are measured, namely through vignettes. Vignettes are better suited for identifying worldviews because they provide a rich holistic formulation that may otherwise not be captured through the sum of a sequence of Likert scales. Specifically, the worldview vignettes offer a flexible approach in which, given a narrative, respondents are allowed to formulate and consider a generalized situational outlook when interpreting them [21]. For instance, when interpreting a *Survivor* worldview, a respondent in India might be despairing about food whilst a respondent in the USA might be despairing about mortgages. Ultimately, despite the differences in personal experiences that respondents draw upon, the psychological experience remains similar. Therefore, worldviews can be thought to be the phenomenological filter for engaging the cognitive miser, acting as a lens through which individuals interpret their own personal experiences.

## Social values

Belief systems and values are linked to each other because the former allows the manifestation of the latter [1, 22]. One could argue that the distinction resembles that between genotype and phenotype in evolution. Values are formulated on the basis of what an individual or social group deems to be important, desirable or favourable, playing a key role in bridging the gap between individual and society [13, 23]. One of the most established theories of values is the one outlined by Schwartz [13]. In his theory Schwartz [13] defines values as individually held subjective beliefs that (a) are strongly associated with feelings (b) refer to desirable goals that motivate action (c) are ordered in level of relative importance, and (d) set a standard on which judgements and decisions are made. Furthermore, values are also defined as universal because they are thought to satisfy three universal requirements of human existence, namely, the needs of individuals as human beings, of harmonious social interaction, and of survival and welfare of social groups [23]. Schwartz's theory of values organizes them in the form of a circumplex consisting of 10 broad value types, namely: (1) self-direction (independent thought and action), (2) stimulation (excitement towards life), (3) hedonism (gratification for oneself), (4) achievement (personal success), (5) power (authority and status), (6) security (safety and stability), (7) conformity (following social norms), (8) tradition (respecting customs), (9)

benevolence (well-meaning towards others), and (10) universalism (respecting of all people and nature) [13, 23]. Empirical evidence for this model emerged from smallest space analysis that examined the spatial relationships amongst the values [13]. Notably, Schwartz et al. [24] have recently developed a more detailed value circumplex consisting of 19 different value types that can, however, be collapsed into the original 10. Additional analysis of the original 10 value circumplex revealed a two-dimensional structure [13, 23]. On the one hand, *Conservation versus Openness to Change* reflects the tension between values relating to preservation or change of the status quo [13]. On the other hand, *Self-Enhancement versus Self-Transcendence* reflects the tension between values relating to personal or other-related interests and successes [13]. Schwartz's theory has been validated across a wide range of countries and cultures, and the measures of these values (i.e., the Schwartz Values Survey [SVS]), has demonstrated strong psychometric properties [24–26]. In addition, meaningful relationships have been reported between values and beliefs [13, 14].

## The hypothesized relationship between worldviews and values

Worldviews and values may be thought of as comprising a hierarchical structure with values being the more abstract and worldviews being the less abstract [27]. Earlier it was noted that values are universal and applicable regardless of context [23]. Particularly, since values transcend specific situations and contexts, they offer an opportunity to understand the motivational constituents that make up one's avowed worldview. The values that we refer to in our study are the higher order values identified by Schwartz [13, 23]. Though Schwartz's value theory and the concept of worldviews are distinct frameworks having different foci and applications, we believe there is a degree of correspondence between the two. Particularly, the two higher order values outlined seemingly correspond with Triandis' dissection of individualism-collectivism dimensions, which have been linked to cultural worldviews [28, 29]. Due to this, Schwartz's higher order value dimensions, that are more individual-oriented, offer an opportunity to link values to our conceptualisation of worldviews. Indeed, with reference to Schwartz's value theory, empirical evidence has provided support for a meaningful relationship between individually-held beliefs and generalized values [30, 31]. For instance, Feldman [30] reported that despite evidence that values and moral foundations are unique and separate constructs, findings still indicate a telling relationship between the two. In one contrast, the *Harm/Care* and *Fairness/Reciprocity* foundations were associated with higher benevolence and universalism values when collapsed under the higher order value of self-transcendence. In the other contrast, the *Loyalty/Betrayal*, *Authority/Respect*, and *Purity/Sanctity* foundations were associated with higher tradition, conformity, and security values when collapsed under the higher order value of conservation [30]. These findings corroborate an earlier meta-analysis examining the value-attitude relationship based on moral foundation theory [32], where self-transcendence values were found to be related to the *Fairness* foundation/pro-environmental attitudes and the *Care* foundation/pro-social attitudes [33]. Conversely, conservation values were found to be related to the *Purity* foundation/religious attitudes and the *Authority* foundation/political attitudes [33]. With reference to research on social axioms and values, *Social Complexity* has been reported to positively correlate with self-direction and benevolence values, *Reward for Application* has positively correlated with conformity values, and *Fate Control* and *Religiosity* have positively correlated with tradition values [31]. Though such studies do not directly tap into the construct of worldviews being investigated here, they are meaningful in their implications on the construct (see Table 1). For this reason, such findings offer a strong basis for predicting a relationship between worldviews and values (see Table 2 for summary of predictions).

Table 2. Summary of worldview descriptions and predictions.

| Worldview | Summary Description | Dominant Supporting Values |
|---|---|---|
| *Localised* | Positive outlook to people and the world, contributing towards the wellness of others. | Self-Transcendence Openness to Change |
| *Orthodox* | Willing to contribute to the wellness of others without the desire to change the status quo. | Self-Transcendence Conservation |
| *Pragmatist* | Willing to bend the rules for loved ones to navigate an unfair world. | Self-Enhancement Openness to Change |
| *Reward* | Hard working with a strong drive for achievement, prioritising in-group over others. | Self-Enhancement Conservation |
| *Survivor* | Fatalistic and cynical view of people and the world. | Conservation |

The *Localised* worldview is associated with a generally positive outlook of people and the world, with a strong willingness to contribute towards the wellness of others. Furthermore, it is also associated with flexible and open views. As outlined in Table 1, this worldview is conceptually linked to the *Social Complexity* social axiom, amongst others. *Social Complexity* has been positively linked to values of self-transcendence [31]. Furthermore, prosocial behaviour, the central underlying characteristic of the *Localised* worldview, has also been linked to values of self-transcendence. In light of these findings, it is reasonable to expect that the *Localised* worldview will correlate positively with values of self-transcendence. Findings relating these beliefs to the values of openness to change, or conservation, are not entirely in synch. Specifically, the *Loyalty/Betrayal* foundation has been linked to values of conservation, however, the *Social Complexity* social axiom has not been linked to either of the values of conservation or openness to change. These noncomplementary findings may be attributed to slight variation in each belief's underlying notions, or perhaps even to differing methodological approaches. Nevertheless, considering that the *Localised* worldview is conceptually linked to open minded-ness, it is expected to positively correlate with values of openness to change. These expected linkages emphasize the significance of other-related interests and the resistance of maintaining a status quo for the *Localised* worldview.

The *Orthodox* worldview is associated with a generally positive outlook of people and the world, however, without the desire to change the status quo. For this reason, this worldview is characterised by rather rigid and convergent thinking, ready to accept the current state of matters with little challenge. Indeed, in a study on views towards recreational cannabis use, Sammut et al. [16] reported that the *Orthodox* worldview stood out from the other worldviews in predicting opposition towards recreational cannabis use. The *Orthodox* worldview is conceptually linked to the *Purity/Sanctity* foundation and the *Religiosity* social axiom, both of which were correlated with values of conservation [30, 31, 33]. Furthermore, *Religiosity* was also found to be positively linked to values of self-transcendence [31]. With consideration to these findings, it is expected that the *Orthodox* worldview correlates positively with values of self-transcendence and conservation.

The *Pragmatist* worldview is associated with distrust in social institutions and a relatively negative outlook of people and the word. Despite this, individuals who endorse the *Pragmatist* worldview also believe that one can easily navigate such a world if one is willing to adapt and bend the rules. This worldview is conceptually linked to the *Fairness/Reciprocity* foundation and *Fate Control* social axiom, amongst others. It is worth noting that findings on the two belief systems are different, namely, the *Fairness/Reciprocity* foundation positively correlates with values of self-transcendence whereas *Fate Control* does not [30, 31, 33]. Another conceptual link to this worldview includes the *Niche of Belongingness* symbolic universe. Salvatore et al. [17] claim that this symbolic universe, along with *Interpersonal Bond*, may be seen as a

source of bonding social capital (i.e., prioritizing in-group identity and cohesion). For this reason, it will be expected that, like *Fate Control* but unlike the *Fairness/Reciprocity* foundation, this worldview will correlate negatively with values of self-transcendence (and so positively with values of self-enhancement). Additionally, due to the element of distrust in those with power, it is expected that the *Pragmatist* worldview will correlate positively with values of openness to change. In an early study investigating the relationship between values and trust in institution, Devos et al. [34] reported that levels of trust correlated positively with values that emphasize security, preservation, and tradition, that is, those values subsumed under the higher order value of conservation. The authors also reported that levels of trust correlated negatively with values that emphasize change and independent action, that is, those values subsumed under the higher order value of openness to change. These findings were later corroborated by Morselli et al. [35] through a multilevel assessment carried out on cross-cultural datasets.

The *Reward* worldview is largely associated with hard work and a strong drive for achievement. Importantly, it is also characterised by obedience and respect of social norms. For this reason, an individual endorsing this worldview believes that life's consequences are generally always fair and deserved, especially if one is unable to exercise restraint over their actions that violate the status quo. The *Reward* worldview is conceptually linked to the *Harm/Care* foundation and the *Reward for Application* social axiom, amongst others. Findings linking the two beliefs to values have shown a positive link between the *Harm/Care* foundation and values of self-transcendence [30, 33], and between the *Reward for Application* social axiom and values of conservation [31]. As outlined earlier, the *Harm/Care* foundation represents the notion of looking after others. Although this aspect is shared with the *Reward* worldview, it is worth noting that possibly, for the *Reward* worldview, caring for others may arise as a by-product of the desire to be in a higher position (i.e., a parental/authority figure). Therefore, it is possible that the *Harm/Care* foundation is rooted in more egalitarian intentions in contrast to the *Reward* worldview. Due to this conceptual difference, it is expected that unlike the *Harm/Care* foundation, the *Reward* worldview positively correlates with values of self-enhancement. Furthermore, due to characteristics relating desire for authority and control to prevent harm, and in line with the findings on the *Reward for Application* social axiom, the *Reward* worldview is expected to correlate positively with values of conservation.

Lastly, the *Survivor* worldview is associated with a fatalistic and cynical view of people and the world. This negative view is also accompanied by significant distrust in society and its institutions. The *Survivor* worldview is conceptually linked to the *Authority/Respect* foundation which was found to correlate positively with values relating to conservation [30]. In addition, it is conceptually linked to the *Social Cynicism* social axiom which has correlated positively with the value of power but not with other values collapsed under the higher order values of self-transcendence and self-enhancement [31]. For this reason, it is not theoretically evident how the *Survivor* worldview correlates with the self-transcendence/self-enhancement value tension. Nevertheless, considering the findings within the domain of moral foundations [30] and findings linking fatalism to the values of conservation [36], it can be reasonably expected that the *Survivor* worldview correlates positively with conservation values.

## Method

This study formed part of a larger exercise investigating the cognitive and behavioural correlates of different beliefs, and their influence on self-regulatory processes. For the present purposes, only methods, data analyses, and results pertaining to the component investigating the correlations between worldviews and values will be reported.

## Participants and procedure

Participants were recruited through Prolific Academic (ProA). ProA has been reported to produce superior data quality for behavioural research when compared to other online recruitment platforms such as Amazon's Mechanical Turk (MTurk) [37]. Since the study was a multi-part study, entailing participation in two different sessions, participants were pre-screened in ProA using the criteria of having already taken part in a minimum of five other studies and possessing an approval rate of 100%. These criteria were selected to ensure that participants had prior experience with using ProA and taking part in online studies.

Participants were provided with an online information letter and consent form outlining the details of the study. Following consent of participation by clicking the "continue to experiment" button, the first session commenced. This session contained the worldviews scale and vignettes, amongst other measures. Once the first session was completed and after a few hours had elapsed, the second session was made available to the same participants. This session contained the PVQ-RR, amongst other sessions. Each session lasted around 20 minutes and participants were rewarded a total of £7.00. This study received self-assessed ethical clearance following the University of Malta's research code of ethics and ethical clearance procedures.

An initial total of 290 participants were recruited, 33 of which failed the attention checks put in place to ensure good quality of the data, and 6 participants failed to participate in the second session, resulting in 251 participants. Out of the 251 participants ($M_{age}$ = 25.12, $SD_{age}$ = 3.20), 156 identified as female and 95 identified as male. All participants resided in one of the OECD countries as per ProA's sign-up criterion. Most participants resided in Europe (66.9%), followed by Africa (17.9%), America and Canada (8.4%), and lastly Australia (6.0%).

## Measures

**Worldviews.**   A list of 5 vignettes were used, each characterising one of the worldviews (see Table 3). The vignettes were created and eventually refined through a preliminary study in which a correlation analysis was carried between items used in the measures of symbolic universes [17], social axioms [18], moral foundations [19], and the initial conceptualisation of the worldview vignettes. For each of these vignettes, participants were asked to rate the extent to which they believed that each vignette applied to themselves using a Likert scale ranging from 1 (not at all) to 5 (completely). In addition, participants were also required to select a single vignette that best approximated their own views. This measure of worldviews has reliably been

**Table 3. Worldview measures.**

| Worldview | Vignette |
|---|---|
| *Localised* | The future depends on us and the choices we make. Every problem has a solution. Each and every one of us can make an effort to fix the laws and institutions so that they can be just and equal for everyone. Like this we can better address the needs of people and society. |
| *Orthodox* | To succeed in life, we need to follow the rules and local customs in order to maintain social order. We also need to show respect to each other and carry out our duties. Like this we can help others in our community. |
| *Pragmatist* | In life we must adapt ourselves to our circumstances and sometimes we need to go with the flow in order to avoid trouble. The rich and powerful protect their own interests, whereas the kind-hearted suffer. Sometimes you have to work around the rules to help your loved ones. |
| *Reward* | In life, you get what you deserve. Life's challenges are overcome with the efforts we make, and these may offer new opportunities. One must co-operate with others, respect authority, and carry out one's duties. Our efforts will eventually lead to success. |
| *Survivor* | In life, things rarely end up well. People are what they are, and good people usually suffer and are exploited. It is best for one to keep his/her head down and get on with it. |

used in a different study that investigated the role of worldviews in predicting support for recreational cannabis use [16].

**Social values.** The Portrait Values Questionnaire (PVQ-RR) is an alternative to The Schwartz Value Survey (SVS) that is more suitable for online administration. It comprises 57 short verbal portraits that describe a person's goals and aspirations, implicitly tapping into a particular value. There are 3 verbal portraits for each of the 19 values. Typically, all portraits are gender-matched with the respondent, however, for the present study, gender neutral pronouns (they/their) were used to facilitate the online administration of the questionnaire. Participants were asked to complete the PVQ-RR by indicating the extent to which they believe they are like the person being described in each of its portraits, using a 5-point Likert scale. A 5-point Likert scale was adopted to match it with other scales being used in the questionnaire. PVQ-RR allowed for the measure of the 10 personal values, originally identified by Schwartz [13], the refined 19 values [24], and 4 higher order values [24]. The questionnaire demonstrates good psychometric properties for measuring personal values in non-clinical groups [24]. Particularly, the mean Cronbach's alpha reliabilities for the 4 higher values in the present sample were: .88 self-transcendence, .79 self-enhancement, .84 openness to change, and .80 conservation.

**Data analyses.** The 19 values, outlined by Schwartz [23], were calculated by taking the average rating across the 3 verbal portraits that are related to the particular value [32]. Following this, the 4 higher order values were calculated as follows: self-transcendence was calculated by computing the mean score of the values of universalism and benevolence; self-enhancement was calculated by computing the mean scores of the values of achievement and power; openness to change was calculated by computing the mean score of the values of self-direction and hedonism; and conservation was calculated by computing the mean score of the value of security, tradition, and conformity [38]. As per the PVQ-RR scoring and analysis instructions recommended by Schwartz [38], centred value scores were calculated to correct for scale use bias. This was especially recommended since the primary modes of analyses for this study was correlation analyses and linear regression. Scores were centred for all values by calculating the mean rating across all items (MRAT) and subtracting this from each of the value scores [38].

To examine differences between worldviews, dummy variables for each worldview were created. Separate dummy regression analysis were then carried out including all the dummy worldviews, with the *Localised* worldview as the reference category and each individual higher order value as the dependent variable. To control for known sex differences and cultural differences in value orientation [39, 40], all analyses entailed a two-block hierarchical model. With reference to cultural differences, since participants resided in a disproportionate variety of countries, these were grouped together in terms of continent. In the [39, 40]two-block hierarchical model, gender (male, female) and continent (Africa, America, Australia, Europe) were included in the first block and the dummy coded worldviews were included in the second block. No issues of collinearity were detected in any of the analyses.

## Results

When asked to select the single worldview that best approximated their own, the majority chose the *Localised* worldview (45.4%), followed by *Pragmatist* (24.7%), *Reward* (13.1%), *Orthodox* (8.8%), and lastly, *Survivor* (8.0%). When asked to rate the extent of their agreement with each individual worldview, the *Localised* worldview received the highest rating ($M = 4.08$, $SD = .79$) whereas the *Survivor* worldview received the lowest rating ($M = 2.60$, $SD = 1.17$). With reference to value orientations, self-transcendence tended to be scored highest by those

preferring the *Localised* worldview ($M$ = .54, $SD$ = .38) and lowest by those preferring the *Reward* worldview. ($M$ = .27, $SD$ = .31). Furthermore, self-enhancement tended to be scored the highest by those preferring the *Reward* worldview ($M$ = -.36, $SD$ = .38) and lowest by those preferring the *Orthodox* worldview ($M$ = -.76, $SD$ = .54). As for openness to change, it tended to be scored the highest by those preferring the *Pragmatist* worldview ($M$ = .37, $SD$ = .32) and lowest by those preferring the Survivor worldview ($M$ = .15, $SD$ = .46). Finally, conservation tended to be scored the highest by those preferring the *Survivor* worldview ($M$ = -.12, $SD$ = .36) and lowest by those preferring the *Localised* worldview ($M$ = 3.27, $SD$ = .54). These findings are illustrated in Fig 1A and 1B.

## Correlations between worldviews and values

An overview of the correlations between the ratings for each individual worldview and the scores for each of the four higher-order values is outlined in Table 4. The *Localised* worldview correlated positively with self-transcendence, $r$ = .17, $p$ = .009, and negatively with self-enhancement, $r$ = -.17, $p$ = .008. In contrast, the *Reward* and *Survivor* worldviews correlated negatively with self-transcendence, $r$ = -.19, $p$ = .003 and self-enhancement, $r$ = -.21, $p$ < .001 respectively. The *Reward* worldview also correlated negatively with openness to change, $r$ = -.15, $p$ = .018, and positively with conservation, $r$ = 18, $p$ = .004. Similarly, the *Orthodox* worldview correlated negatively with openness to change, $r$ = -.15, $p$ = .019, and positively with conservation, $r$ = 19 $p$ = .003. Lastly, the *Pragmatist* worldview was the only worldview to correlate negatively with conservation $r$ = -.13, $p$ = .040.

## Regression analyses

To investigate the influence of worldviews on higher order values (see Fig 1A and 1B), a series of hierarchical regression models, with each of the four higher order values as the dependent variable, was carried out. As noted earlier, the first block of the model analysed the influence of the demographic variables (gender, continent) whereas the second block analysed the influence of worldviews.

The model predicting self-transcendence was significant, $R^2$ = .186, $F(9, 241)$ = 3.719, $p$ < .001. In this model, the worldviews alone contributed to 11.5% of the variance. Relative to the *Localised* worldview, all but the *Orthodox* worldview resulted in lower self-transcendence: the *Reward*, β = -.261, $t(241)$ = -4.037, $p$ < .001, *Survivor*, β = -.181, $t(241)$ = -2.191, $p$ = .029, and *Pragmatist* worldview, β = -.263, $t(241)$ = -5.072, $p$ < .001, predicted lower self-transcendence scores.

The model predicting self-enhancement was significant, $R^2$ = .96, $F(9, 241)$ = 2.844, $p$ < .01. In this model, the worldviews alone contributed to 6.7% of the variance. Relative to the *Localised* worldview, the *Reward*, β = .302, $t(241)$ = 2.834, $p$ = .005, and *Pragmatist* worldview, β = .283, $t(241)$ = 3.304, $p$ < .001, predicted higher self-enhancement scores.

The model predicting openness to change was significant, $R^2$ = .112, $F(9, 241)$ = 3.372, $p$ < .001. In this model, the worldviews alone contributed to 4.6% of the variance. Relative to the *Localised* worldview, the *Reward*, β = -.167, $t(241)$ = -2.272, $p$ = .024, and *Survivor* worldview, β = -.228, $t(241)$ = -2.418, $p$ = .016, predicted lower openness to change scores.

Lastly, the model predicting conservation was also significant, $R^2$ = .131, $F(9, 241)$ = 4.030, $p$ < .001. In this model, the worldviews alone contributed to 4.3% of the variance. Relative to the *Localised* worldview, the *Reward*, β = .160, $t(241)$ = 2.310, $p$ = .022, *Survivor* worldview, β = .201, $t(241)$ = 2.258, $p$ = .025, and *Orthodox* worldview β = .191, $t(241)$ = 2.326, $p$ = .021, predicted higher conservation scores.

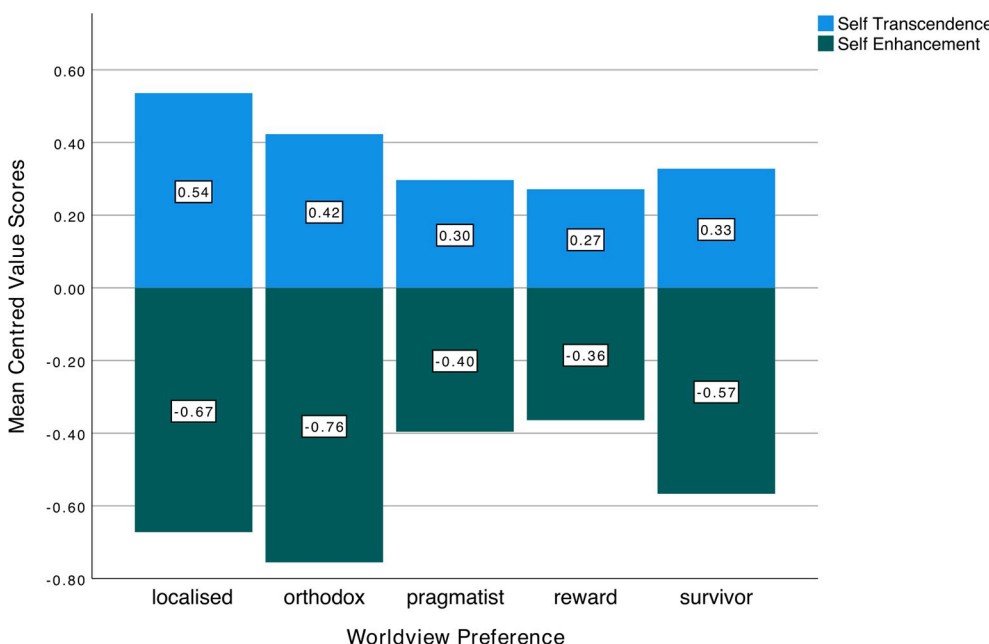

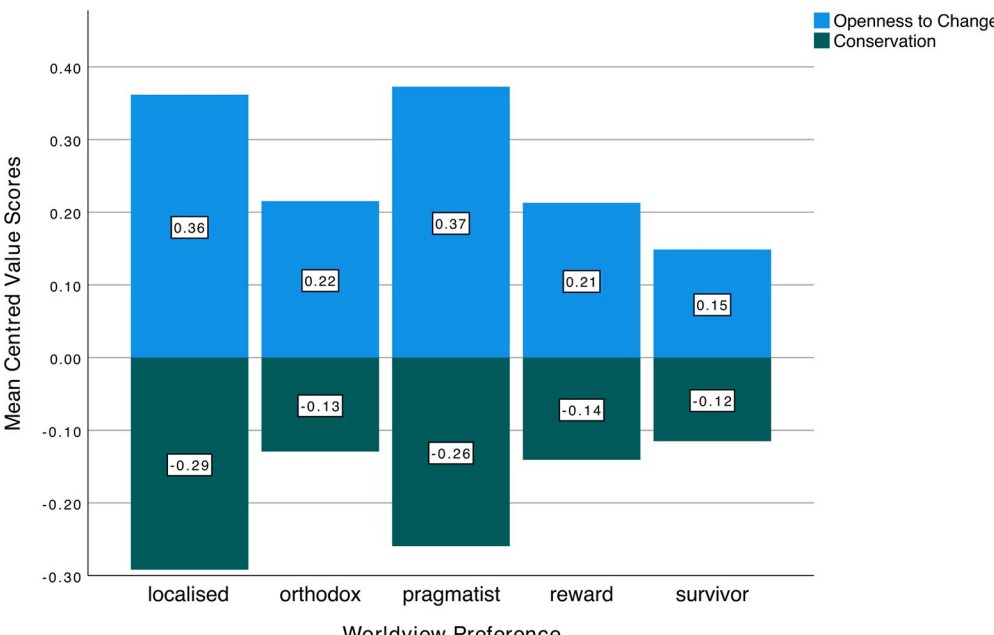

**Fig 1.** A. Mean (centred) score for self-transcendence and self-enhancement grouped by worldview. B. Mean (centred) score for openness to change and conservation grouped by worldview.

## Discussion

The present paper examined the relationship between worldviews and values. Correlational analysis and a series of hierarchical linear regressions were carried out to assess this relationship as well as the influence that worldviews exert on specific higher order values. The findings show that, even though some belief systems, referred to herein as worldviews, are evidently

**Table 4. Correlations between higher order values and worldviews.**

| Variable | 1 | 2 | 3 | 4 | 5 | 6 | 7 | 8 | 9 |
|---|---|---|---|---|---|---|---|---|---|
| 1. Self-Transcendence [a] | - | | | | | | | | |
| 2. Self-Enhancement [a] | -.627** | - | | | | | | | |
| 3. Openness to Change [a] | -.022 | .106 | - | | | | | | |
| 4. Conservation [a] | -.290** | -.308** | -.668** | - | | | | | |
| 5. Localised | .165** | -.166** | -.018 | .039 | - | | | | |
| 6. Pragmatist | -.030 | .084 | .112 | -.130* | .058 | - | | | |
| 7. Orthodox | -.033 | -.108 | -.148* | .188** | .226** | .057 | - | | |
| 8. Reward | -.189** | .104 | -.149* | .180* | .186** | -.005 | .292** | - | |
| 9. Survivor | -.208** | .122 | -.107 | .077 | -.066 | .204** | -.084 | -.005 | - |
| *M* | .416 | -.562 | .315 | -.236 | 4.08 | 3.72 | 3.42 | 2.89 | 2.60 |
| *SD* | .354 | .555 | .388 | .369 | .794 | .985 | 1.026 | 1.220 | 1.174 |

[a] Centred value scores

* $p < .05$.

** $p < .01$.

different from each other, there are nevertheless unique points of convergence that may notably be attributed to underlying values. Moreover, when compared to the results of the regression analyses, the weaker results from the correlation analyses lend further support to the notion that, in quantitative research, the worldviews construct is best suited for explaining a proportion of variance that may otherwise remain unaccounted for by the predictor variables [10].

The *Localised* and *Orthodox* worldviews both agree on the value of self-transcendence but disagree on the value of conservation. Therefore, an individual who endorses either of these two worldviews is likely to be someone who subscribes to an egalitarian view of the world motivated to go beyond selfish desires to help and connect with others. However, what differentiates these two worldviews is the extent to which one is willing to act autonomously and freely. Those who endorse a *Localised* worldview are open to independence and are unrestricted by the need to abide by social order, whereas those who endorse an *Orthodox* worldview are more self-restricting and more comfortable acting within the confines of tradition and society. Though not empirically investigated, this difference between the two might be attributed to the sense of religiosity or belief in higher supremacy that characterises the *Orthodox* worldview. Earlier, the *Orthodox* worldview was proposed to be conceptually linked to the *Ordered Universe* symbolic universe, the *Religiosity* social axiom, the *Purity/Sanctity* moral foundation, and the *Worshipper* deep story profile. Notably, these have all been described as involving an underlying religious notion [17–20]. It could, therefore, be the case that the *Orthodox* worldview is linked to conservatism due to the tendency to adhere to religious teachings and the security that comes with that, undermining an element of agency and self-direction.

Like the *Orthodox* worldview, the *Reward* and *Survivor* worldviews also value conservation. The *Reward* worldview has been conceptually linked to the *caring society* symbolic universe, the *Reward for Application* social axioms, the *Harm/Care* moral foundation, and the *Cosmopolitan* deep story profile. A common feature underlying these beliefs is the importance of forming coalitions, developing trust, and living peacefully with others [17–20]. A reason for the link between the *Reward* worldview and conservation may arise out of the desire of maintaining peace within one's group. For this worldview, it is possible that such peace is thought to be best achieved by exercising control and establishing and adhering to group norms. On a different

note, the *Survivor* worldview, that has been conceptually linked to fatalistic and cynical beliefs, may be linked to conservation because of a sense of powerlessness. That is, even though one is distrustful of society, one would rather let matters remain as they are rather than risk having to adapt to something new. Put simply, for the *Survivor* worldview, "it is better the devil you know than the angel you do not know". Unlike the *Orthodox* worldview, the *Reward* and *Survivor* worldviews do not value self-transcendence. Rather, the *Reward* worldview, in particular, has been linked to self-enhancement. Self-enhancement represents personal focus and self-protection. However, for the *Reward* worldview, the positive link with self-enhancement is not necessarily solely highlighting self-serving motives but could, more fittingly, be highlighting the importance of ingroup unity over outgroup helping. Therefore, for this worldview, ingroup favouritism may also explain a positive link with self-enhancement. The *Survivor* worldview is associated with cynicism. Early studies found empirical evidence linking cynicism with lower self-esteem and lower levels of interpersonal trust [41]. Such negative portrayals of the self and others may be a possible cause that explains why the *Survivor* worldview devalues self-transcendence and is not particularly linked to self-enhancement.

Similar to the *Reward* worldview, the *Pragmatist* worldview also has a negative relationship with self-transcendence and a positive relationship with self-enhancement. The *Pragmatist* worldview was earlier conceptually linked to *Niche of Belongingness* symbolic universe, the *Fate Control* social axioms, the *Fairness/Reciprocity* moral foundation, and the *Rebel with a cause* deep story profile. An underlying theme of these beliefs is a preference for individual autonomy coupled with reciprocal favouritism [17–20]. Essentially, the *Pragmatist* worldview utilises the "tit-for-tat" strategy to navigate the world. This strategy, which is synonymous with reciprocal altruism [42], is based on the principle that one reciprocates the other's actions, collaborating only with individuals who are willing to return the favour [43]. The "tit-for-tat" strategy is an essential survival mechanism because it helps to protect self-interest whilst living peacefully with others [43]. It could be the case that the *Pragmatist* worldview is linked with self-enhancement because their actions are primarily driven by selfish intentions despite seeming to be altruistic in nature. The *Pragmatist* worldview potentially presents itself as a good example of how, ultimately, reciprocal altruism is rooted in a self-serving agenda [44].

The conceptual link between the *Pragmatist* worldview and openness to change differentiates it from the *Reward* worldview. A reason for this could be that individuals who endorse the *Reward* worldview find security in their social group whereas those with a *Pragmatist* worldview do not. Earlier, the *Pragmatist* worldview was related to a negative view of people and society, making them less likely to depend on others. This in turn makes individuals who endorse this worldview more likely to think and act independently, offering an explanation as to why one would be less willing to act within societal constraints.

## Conclusion and future directions

The objective of this inquiry constitutes a starting point for understanding how worldviews may play a role in the formation of coalitions for action [8]. Specifically, the empirical relationship between worldviews and values facilitates the understanding of how individuals may come together and agree to support a cause or a course of action despite clear and widespread intra-group differences. In the pursuit of any cause, some stand to agree for one reason whereas others may agree with the cause or ends pursued for quite different reasons. We propose, therefore, that such agreement involves the coalition of worldviews. In other words, a worldview can ally with another worldview in the pursuit of conservative projects. This would be the case, for instance, in an alliance forged between those holding a *Reward* worldview and others holding an *Orthodox* worldview. That coalition, however, may well crumble should self-

enhancing versus self-transcendent projects rise to the fore, at which point, the *Reward* world-view will find an ally in *Pragmatist* worldviews whilst the *Orthodox* emerge as a common opponent. This simple example illustrates the potential of understanding worldviews in explaining shifting coalitional dynamics in contemporary political landscapes.

A second domain of inquiry that requires empirical effort concerns the endorsement of worldviews and their cognitive correlates. A key question that arises in this theoretical formulation is whether worldviews are marked by individual differences in cognition that result out of inherent dispositions that incline some individuals towards a worldview more strongly than others, or whether, as Sammut [15] proposed, the worldview repertoire is accessible to all individuals with its utility exclusively contingent on situational circumstances. Sammut [15] proposes that individuals are able to change worldviews to ensure adaptation should their life conditions change. In this way, an individual pursuing a *Reward* worldview may, following a series of unfortunate events, emerge with a *Survivor* worldview that enables that individual to face adversity with grit even though there may be little to no personal gain. In essence, human subjects equipped with more or less similar cognitive power or prowess, as it were, should not be inclined one way or another relative to any particular worldview. Such inclinations should accrue solely as a consequence of life circumstances. Empirical study is required to determine whether this is indeed the case or whether, by contrast, the endorsement of worldviews is underlined by individual differences in cognition that incline people in determined directions.

A third domain of inquiry concerns the influence of educational attainment on one's worldview. A higher level of education is known to act as a catalyst for expanding knowledge, engaging in critical thinking [45], increasing tolerance towards diverse others [46], and facilitating political and civic engagement [47]. Owing to this, for instance, one might expect that a higher level of education may predispose individuals towards a *Localised* worldview. Individuals with a higher level of education may be more motivated to address social issues due to the fact that they are exposed to diverse perspectives and are aware of the range of social issues that may accompany them. It would be worth exploring whether this is the case and, if so, to what extent does education play a role in worldviews when compared to other factors such as socioeconomic background.

A final domain of inquiry that emerges from the above concerns is how malleable the endorsement of worldviews might be over time. Developmentally, some people face certain circumstances at birth that may be markedly different from those faced by different others, predisposing them to a particular worldview over another. Consequently, one wonders whether worldviews change in the face of changing life circumstances and what processes govern such adaptation. For instance, one could determine whether adverse life events like divorce or job loss could nudge individuals towards a *Survivor* worldview. Developmental/longitudinal research is required to potentially inform the helping professions and their ability to prescribe psychological remedies in the form of changing outlooks in line with more adaptive worldviews considering the individual's own circumstances. In this light, it will be worth looking at the role played by certain demographics in the endorsement of worldviews and the extent to which this endorsement may be a function of grand ecological circumstances that mark generational eras. For instance, the study of worldviews stands to be informative in understanding differences between pre- to post-Covid mentalities that may go on to mark the perceptions and expectations of generations to come.

## Supporting information

**S1 Data.**
(SAV)

## Author Contributions

**Conceptualization:** Rebekah Mifsud.

**Formal analysis:** Rebekah Mifsud.

**Methodology:** Rebekah Mifsud.

**Writing – original draft:** Rebekah Mifsud.

**Writing – review & editing:** Rebekah Mifsud, Gordon Sammut.

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
