## [Decision Letter · Decision Letter 0]

13 Jun 2023

PONE-D-23-05362Worldviews and the role of social values that underlie themPLOS ONE

Dear Dr. Mifsud,

Thank you for submitting your manuscript to PLOS ONE. After careful consideration, we feel that it has merit but does not fully meet PLOS ONE’s publication criteria as it currently stands. Therefore, we invite you to submit a revised version of the manuscript that addresses the points raised during the review process. First allow me to apologize for the delay. I struggled to find qualified and willing reviewers for this paper. Second (and thankfully) they both felt the paper is nearly ready, just requiring some minor revisions. Please submit your revised manuscript by Jul 28 2023 11:59PM. If you will need more time than this to complete your revisions, please reply to this message or contact the journal office at plosone@plos.org. Please include the following items when submitting your revised manuscript:A rebuttal letter that responds to each point raised by the academic editor and reviewer(s). You should upload this letter as a separate file labeled 'Response to Reviewers'.A marked-up copy of your manuscript that highlights changes made to the original version. You should upload this as a separate file labeled 'Revised Manuscript with Track Changes'.An unmarked version of your revised paper without tracked changes. You should upload this as a separate file labeled 'Manuscript'.If applicable, we recommend that you deposit your laboratory protocols in protocols.io to enhance the reproducibility of your results. Protocols.io assigns your protocol its own identifier (DOI) so that it can be cited independently in the future. For instructions see: https://journals.plos.org/plosone/s/submission-guidelines#loc-laboratory-protocols. Additionally, PLOS ONE offers an option for publishing peer-reviewed Lab Protocol articles, which describe protocols hosted on protocols.io. Read more information on sharing protocols at https://plos.org/protocols?utm_medium=editorial-email&utm_source=authorletters&utm_campaign=protocols.

We look forward to receiving your revised manuscript.

Kind regards,

Peter Karl Jonason

Academic Editor

PLOS ONE

Reviewers' comments:

Reviewer's Responses to Questions

**Comments to the Author**

1. Is the manuscript technically sound, and do the data support the conclusions?

Reviewer #1: Yes

Reviewer #2: Yes

2. Has the statistical analysis been performed appropriately and rigorously? 

Reviewer #1: Yes

Reviewer #2: Yes

3. Have the authors made all data underlying the findings in their manuscript fully available?

Reviewer #1: Yes

Reviewer #2: Yes

4. Is the manuscript presented in an intelligible fashion and written in standard English?

Reviewer #1: Yes

Reviewer #2: Yes

5. Review Comments to the Author

Reviewer #1: This was a very interesting paper that contributes to the theorisation and empirical study of socio-cognitive mechanisms of worldviews/values. It was wonderful to see that you try to advance a more situational understanding of the concept, which is lacking in the field. I have some minor comments that I hope can be helpful to improver your wonderful work.

1. Even though you clarify the similarities between worldviews and the other concepts are, it is not very clear what are the differences between them and how the idea of the worldview is a better choice. Is it possible to explain a bit more?

2. You are highlighting the importance of situational demands in the conceptualisation and operationalisation of worldviews. Yet it is not entirely clear how this is reflected in the operationalisation of the construct i.e. in the vignettes. For example, who are these ‘others’ in the scenarios, who constitutes an in-group/out-group can vary across situations and topics/projects of interest. You introduce this idea towards the conclusion but it is important to clarify what situational aspects (if any) the vignettes are taking into account. Another point is that you are drawing on Schwartz’s values which are quite abstract and do not take into account the importance of situations and contexts. Can you please explain address this?

3. Can you please explain a bit more why you chose for your study Schwartz’s values as opposed to the other constructs you review in your introduction?

4. In the sampling section, can you please provide some more information about the educational background of the participants? Did you also control for that in your analyses? If not, it's ok. Can you just please explain why.

5. In the results section, some of the correlations are quite weak. Can you please comment on that and what implications this has for your theorisations and findings?

6. In the results section, why did you choose to perform the analyses with the higher order values eg self-transcendence and not the individual ones e.g. benevolence, universalism ? There has been some evidence that goes against Schwartz’s value relationships. Can you explain your rationale?

7. In terms of further research and reflections, the vignettes seem to capture rather secular and anthropocentric worldviews. What about relationships to more than human worlds eg nature, spirits, but also orientations to time (role of temporality- how we understand history links to our worldviews maybe Lola Olufemi could be an inspiration?). Another question is: Does a worldview include what is or also what could be different? Perhaps it would be interesting to engage with questions about speculation and world-making e.g. Power, S. A., Zittoun, T., Akkerman, S., Wagoner, B., Cabra, M., Cornish, F., Hawlina, H., Heasman, B., Mahendran, K., Psaltis, C., Rajala, A., Veale, A., & Gillespie, A. (2023). Social Psychology of and for World-Making. Personality and Social Psychology Review, 0(0). https://doi.org/10.1177/10888683221145756

8. In your conclusion, you argue about the role of situations and you bring the example of poor versus rich people. You argue ‘poverty may predispose individuals towards a Survivor worldview whilst riches might predispose others towards a Reward worldview’ Actually this is an erroneous belief. There is a lot of evidence to the contrary. You can refer to literature on development eg Robert Chambers, Arturo Escobar

Congratulations on a super interesting paper and for making a case for the importance of situations in the study of worldviews.

Reviewer #2: please see attached document--------------------------------------------------------------------------------------------------------------------------------------------------------------------------------------------------------------------------

6. PLOS authors have the option to publish the peer review history of their article (what does this mean?). If published, this will include your full peer review and any attached files.

Reviewer #1: No

Reviewer #2: No

---

## [Author Response · Author response to Decision Letter 0]

27 Jun 2023

Specific responses have been provided in an uploaded letter titled "Response to Reviewers"

---

## [Editor Report · Decision Letter 1]

28 Jun 2023

Worldviews and the role of social values that underlie them

PONE-D-23-05362R1

Dear Dr. Mifsud,

We’re pleased to inform you that your manuscript has been judged scientifically suitable for publication and will be formally accepted for publication once it meets all outstanding technical requirements.

Kind regards,

Peter Karl Jonason

Academic Editor

PLOS ONE
---

## [Editor Report · Acceptance letter]

3 Jul 2023

PONE-D-23-05362R1 

Worldviews and the role of social values that underlie them 

Dear Dr. Mifsud:

I'm pleased to inform you that your manuscript has been deemed suitable for publication in PLOS ONE. Congratulations! Your manuscript is now with our production department. 

Kind regards, 

on behalf of

Dr. Peter Karl Jonason 

Academic Editor

PLOS ONE